# Impact of COVID-19 Pandemic on Children Visiting Emergency Department for Mental Illness: A Multicenter Database Analysis from Korea

**DOI:** 10.3390/children9081208

**Published:** 2022-08-11

**Authors:** Woori Bae, Arum Choi, Seonjeong Byun, Kyunghoon Kim, Sukil Kim

**Affiliations:** 1Department of Emergency Medicine, College of Medicine, The Catholic University of Korea, Seoul 06591, Korea; 2Department of Preventive Medicine and Public Health, College of Medicine, The Catholic University of Korea, Seoul 06591, Korea; 3Department of Neuropsychiatry, Uijeongbu St. Mary’s Hospital, College of Medicine, The Catholic University of Korea, 271, Cheonbo-Ro, Uijeongbu-si 11765, Korea; 4Department of Pediatrics, Seoul National University College of Medicine, Seoul 13620, Korea

**Keywords:** COVID-19, child, emergency department, mental illness

## Abstract

We aimed to identify changes in the proportion of pediatric emergency department (PED) visits due to mental illness during the coronavirus disease 2019 (COVID-19) pandemic. This was a retrospective observational study of visits to the PED at six university hospitals from January 2017 to December 2020. We included children aged 5–17 years who were diagnosed with a mental illness. We used segmented regression analysis to identify the change in the proportion of patients with mental illness. A total of 845 patients were included in the analysis. After the first case of COVID-19 was reported in Korea, the number of PED visits significantly decreased by 560.8 patients per week (95% confidence interval (CI): −665.3 to −456.3, *p* < 0.001). However, the proportion of patients with mental illness increased significantly, by 0.37% per week (95% CI: 0.04% to 0.70%, *p* = 0.03), at this time point. Subgroup analyses revealed that emotional disorders significantly increased by 0.06% per month (95% CI: 0.02% to 0.09%, *p* < 0.001) during the pandemic. Our study revealed that an increased proportion of patients with mental illness visited the PED during the COVID-19 pandemic. Specifically, we identified that the proportion of emotional disorders continues to rise during this pandemic.

## 1. Introduction

In 2019, the coronavirus disease 2019 (COVID-19) began in Wuhan, China, and spread worldwide [1]. The World Health Organization declared the COVID-19 pandemic on 11 March 2020, and as of 2 August 2021, there have been 197,882,153 COVID-19 confirmed cases and 4,219,951 deaths in 223 countries worldwide [2]. To minimize the spread of COVID-19, each country’s government has implemented social distancing [3]. This reduced people’s gatherings and communications and increased the amount of time alone. This has led to an increase in patients complaining of emotional disorders such as depression, anxiety, and stress. According to a previous study, the prevalence of depression, anxiety, and stress in adolescents during the COVID-19 pandemic was 33.7%, 31.9%, and 29.6%, respectively [4]. This is much higher than that reported by a previous report of a lifetime prevalence of depressive symptoms of 6.8% [5]. A previous study in Korea also reported that during the pandemic, the proportion of patients visiting the emergency department for psychosis and mood disorders increased by 12% and 21%, respectively [6]. In addition, a longitudinal study reported that the proportion of children and adolescents with anxiety and depression during the COVID-19 pandemic significantly increased from 14.9% to 30.1% and 10.0% to 15.1%, respectively, compared to before the COVID-19 pandemic [7].

The COVID-19 pandemic has prolonged, and the period of implementation of social distancing has been extended. Moreover, as the reopening of schools is delayed and outside activities are refrained, communication between children has decreased, and the time spent alone has increased [8]. As a result, children feel anxious and lonely, leading to psychologically unhealthy lives [9]. Eventually, these situations can lead to mental illnesses, such as suicide attempts and drug use [10]. However, despite these implications, there are few studies on this in pediatric patients.

We hypothesized that a prolonged COVID-19 pandemic resulted in increased pediatric emergency department (PED) visits in children with mental illness. This study aimed to determine whether the number and proportion of children with mental illness who visited PEDs increased during the pandemic.

## 2. Materials and Methods

### 2.1. Study Design and Setting

This study is a retrospective observational study conducted on patients who visited the PED at the six university hospitals in Korea between 2017 and 2020. We obtained and analyzed all the data from the electronic medical records (EMR) of hospitals.

### 2.2. Selection of Participants

We included children aged < 18 years who visited the PED of the six hospitals during the study period. Those who visited the PED for issuance of a certificate or were discharged from the PED with R-codes and those without proper data were excluded. An issuance of a certificate refers to a patient who visited the PED to receive a document proving that he or she received treatment. An R-code is a code starting with R in ICD-10 and indicates symptoms, signs, or abnormal findings that are not classified otherwise. Most children under the age of 5 years visited the PED for substance use but were excluded because they were not intentionally taking the substance. Therefore, we analyzed patients aged 5–17 years who visited the PED for mental illness.

### 2.3. Measurements

We considered an independent case of PED visits from patient arrival to discharge. We investigated the following variables in the EMR of each hospital: demographic variables including age and sex, acuity of patient’s status at arrival, diagnoses, and disposition. The acuity of patient’s status at arrival was evaluated using the Korean Triage and Acuity Scale (KTAS) [11]. The KTAS, which was made by modifying the Canadian Triage and Acuity Scale [12] to suit the Korean situation, classifies the acuity of a patient’s status into level 1 (critical) to 5 (non-urgent). Diagnoses were assigned to the patient based on the diagnostic codes of the Korean Standard Classification of Diseases-7 [13], the Korean version of the International Classification of Diseases-10th Revision (ICD-10) [14]. A patient with mental illness was defined as a patient whose main diagnostic code was the mental health problem code in the ICD-10. We classified mental illness as anxiety disorders, mood disorders, somatoform disorders, developmental disorders, schizophrenia and other psychotic disorders, alcohol-related disorders, substance-related disorders, intentional self-harm, and miscellaneous. The mental illness codes of the ICD-10 are indicated in Appendix A.

### 2.4. Patient and Public Involvement

There was no patient or public involvement in setting the research agenda.

### 2.5. Statistical Analysis

Differences in age, sex, KTAS level, arrival mode, and disposition were analyzed using the chi-squared test. We used segmented regression analysis to identify the change in the proportion of patients with mental illness by week. Data were adjusted for seasonality because there was seasonal variation in patients visiting the PED.

We performed all analyses using R version 4.0.0 (R Foundation for Statistical Computing, Vienna, Austria), with the probability level for significance set at a *p*-value < 0.05.

## 3. Results

A total of 257,835 children aged < 18 years visited the PED at the six hospitals from January 2017 to December 2020. Of these, 1923 cases who visited the PED to obtain a certificate, and 9484 cases with R-codes as the diagnostic code, were excluded. A further 1358 cases without diagnostic codes and 48 cases without arrival records were excluded. Finally, 244,177 cases with no mental illness code and 498 cases of patients aged < 5 years were excluded, and then 845 patients were included in the analysis (Figure 1).

During the COVID-19 pandemic in 2020 and the 3 years preceding it, there were 845 children with mental illness aged 5–17 years. Of these, 188 were diagnosed with mental illness in 2020, and the remaining 657 were diagnosed in 2017 through 2019. The proportion of children with mental illness among all children who visited the PED significantly increased from 0.77% in 2017 through 2019, to 1.26% in 2020 (*p* < 0.001). The mean number of weekly visits decreased to 3.6 in 2020 compared to the previous mean of 4.2 (*p* = 0.051). Compared to the past three years, in the first year of the COVID-19 pandemic, the proportion of visits for the 10–14 years age group decreased, while the 15–17 years age group increased in contrast. The proportion of females was similar (68.49% vs. 69.15%). Other demographic data are provided in Table 1.

The segmented regression analysis showed a change in PED visits as of the fourth week of January 2020, when the first patient of COVID-19 was announced in Korea. At this point, the number of PED visits significantly decreased by 560.8 patients per week (95% confidence interval (CI): −665.3 to –456.3, *p* < 0.001). Subsequently, the total number of PED visits fell by an additional 4.3 patients per week (95% CI: −8.0 to –0.6, *p* = 0.022) (Figure 2A). Conversely, the proportion of patients with mental illness increased significantly by 0.37% per week (95% CI: 0.04–0.70%, *p* = 0.03) during this period. There was no evidence of a significant change in trend during the COVID-19 pandemic in the proportion of patients with mental illness, however, it increased by 0.01% per week (95% CI: −0.01–0.01%, *p* = 0.48) (Figure 2B).

Since the type and frequency of mental illness are different across age and sex, we divided the patients into subgroups and performed segmentation regression analysis. There were no significant trends in the proportion of both males and females during the COVID-19 pandemic, whereas the proportion of females with mental illness significantly increased by 0.03% per month (95% CI: 0.01–0.05%, *p* < 0.001) before the COVID-19 pandemic. When patients with mental illness were divided into subgroups according to age, there were no significant trends in the proportion of patients in both age groups (10–14 years and 15–17 years) during the COVID-19 pandemic. In contrast, before the COVID-19 pandemic, the proportion of patients in both age groups was significantly increased by 0.01% and 0.06%, respectively (95% CI: 0.00–0.02%, *p* = 0.001, and 95% CI: 0.02–0.09%, *p* < 0.001, respectively) (Figure 3A–D).

Substance-related disorders are the most common mental illness in patients visiting the PED, followed by anxiety, somatoform, alcohol-related, and mood disorders. Substance-related, anxiety, somatoform, and mood disorders were 0.31%, 0.12%, 0.09%, and 0.05%, respectively, among all PED visits before the COVID-19 pandemic, and significantly increased to 0.53%, 0.22%, 0.16%, and 0.11%, respectively, during the COVID-19 pandemic (*p* < 0.001, *p* = 0.003, *p* = 0.006, and *p* = 0.001, respectively) (Table 2).

Alcohol- and substance-related disorders were categorized as substance use disorders and analyzed. There was no significant trend in the proportion of substance use disorders during the COVID-19 pandemic, whereas it significantly increased by 0.01% per month (95% CI: 0.00–0.02%, *p* < 0.001) before the COVID-19 pandemic. Anxiety, somatoform, and mood disorders were categorized as emotional disorders and analyzed. There was no significant trend in the proportion of emotional disorders before the COVID-19 pandemic, whereas it increased significantly by 0.06% per month (95% CI: 0.02–0.09%, *p* < 0.001) during the COVID-19 pandemic (Figure 3E,F).

## 4. Discussion

This study showed that the proportion of total PED visits for mental illness in Korea increased during the COVID-19 pandemic compared to that before the pandemic. Among them, the increase in the proportion of patients with anxiety disorders, mood disorders, somatoform disorders, and substance-related disorders was remarkable. Additionally, it has been identified that the number of patients with emotional disorders continues to increase during the COVID-19 pandemic.

The strength of our study is that it was analyzed using a large dataset extracted from six university hospitals. In addition, the six university hospitals are located in the metropolitan area and provinces where more than 50% of the total population of Korea resides. Therefore, selection and information biases were minimized by analyzing the large dataset. Moreover, we collected data for one year after the first case of COVID-19 in Korea and adjusted for seasonal variations of PED visits.

Studies have shown that the total number of patients visiting the PED decreased after the onset of the COVID-19 pandemic [15,16]. This is consistent with the results of our study. However, our study showed that although the number of patients with mental illness visiting the PED has decreased, their proportion has increased relatively since the onset of the COVID-19 pandemic. A previous study also reported that compared to 4.0% before the COVID-19 pandemic, the proportion of patients visiting the PED for mental illness has significantly increased to 5.7% after the onset of COVID-19 [17]. This may be because of the disconnection of communication among adolescents during the COVID-19 pandemic and the resulting friction between family members. As the reopening of schools was delayed and outdoor activities decreased, children and adolescents had reduced communication with their peers, and the longer time they spent at home, the more conflicts they had with family members [18]. Moreover, several studies have reported an increase in anxiety, depression, and post-traumatic symptoms among children and adolescents due to the disconnection of communication and friction between family members during the COVID-19 pandemic [19,20,21]. Another reason is thought to be a decrease in non-mental illnesses. Due to social distancing and strict personal hygiene during the COVID-19 pandemic, the prevalence of infectious diseases, particularly respiratory infections, have decreased. A study in Singapore reported that respiratory infections were significantly reduced during the lockdown due to the COVID-19 pandemic compared to the pre-lockdown period [22]. Additionally, due to limited outdoor engagement and activities, the number of injured patients has also decreased. An Irish study reported the fewest traumatic hospitalizations for children during the COVID-19 pandemic over the past decade [23]. Restrictions on access to primary care may be another reason. Due to social distancing during the COVID-19 pandemic, outdoor activities or gatherings were restricted, which made it difficult to use primary clinics and counseling centers. Therefore, when a patient with a mental illness is experiencing psychological distress, the patient and their caregiver have no choice but to visit the PED, which may increase the proportion of patients with mental illness.

We found that there was no significant change in the trend of PED visits in both male and female patients during the COVID-19 pandemic. However, prior to the pandemic, the proportion of female patients was significantly increasing. In the United States, a study on the trends in the emergency department visits due to psychological problems in patients aged 6–24 years from 2011 to 2015 showed an insignificant increase in male patients from 8.9% to 11.0%, but a significant increase from 6.9% to 9.5% in female patients [24]. Other studies in the United States reported an overall increase in the prevalence of depression and suicide among adolescents, with a more pronounced increase in females than in males [25,26]. This is probably because females experience more frequent parental separation and divorce, challenges, and violence at home during childhood than males [27].

Adolescence is a particularly vulnerable period for psychological problems [28], and about half of mental illnesses develop after the age of 14 [29]. Therefore, we divided the study patients into a 10–14 years age group and a 15–17 years age group. There was no significant trend in both groups during the COVID-19 pandemic; however, there was a significantly increasing trend in the proportion of mental illness in both groups prior to the pandemic. A WHO report found that health-threatening behaviors were mildly frequent in the younger age group of 11 years, whereas health-threatening behaviors increased with age in 13- and 15-year-olds [30]. Another study reported higher rates of suicide in older adolescent age groups [31]. Suicide attempts in younger children are often impulsive, whereas, in older adolescents, it may be related to stress, self-doubt, pressure, and loss. Thus, appropriate measures will be required to account for such differences with respect to age group.

Based on the diagnosis of mental illness, we identified that the proportion of anxiety disorders, mood disorders, somatoform disorders, and substance-related disorders increased significantly during the COVID-19 pandemic. Stressful environments and events can be triggers for mental illness, particularly depression and anxiety [32,33,34]. The COVID-19 pandemic is an unpredictable and stressful event worldwide. Additionally, children and adolescents are more vulnerable to stressful events than adults. During the COVID-19 pandemic, approximately 20% of adults reportedly met the criteria for one or more psychological problems, in comparison to more than 33% of children and adolescents [35]. In children and adolescents, the symptoms of depression and anxiety are so diverse that they are often unrecognized and untreated, eventually leading to emotional disorders, such as anxiety disorders, mood disorders, and somatoform disorders [36]. Moreover, we identified an increase in the proportion of substance-related disorders during the COVID-19 pandemic. Restrictions of outdoor activities and friction with family can heighten stress, which may lead to emotional confusion and impulsivity in adolescents, which may ultimately lead to substance abuse [37,38].

There were some limitations to this study. First, due to the retrospective and observational nature of the study, there is a possibility of selection and information biases. However, these biases were minimized by analyzing a large set of data. Second, since we received and analyzed anonymized patient data, there is a possibility that the same patient’s information was included more than once. This is an inherent limitation of studies with anonymized patient data and is not specific to this study. Finally, we could not identify the patient’s prior history of mental illness. Therefore, in this study, it was not possible to differentiate between patients with a first diagnosed mental illness and patients with a preexisting mental illness that worsened.

## 5. Conclusions

This study revealed that an increased proportion of patients with mental illness visited the PED during the COVID-19 pandemic. Specifically, we identified that the proportion of emotional disorders continues to rise during the COVID-19 pandemic. Therefore, as the COVID-19 pandemic persists, there is a need to develop long-term planning to provide appropriate first aid to patients with mental illness visiting the PED.

## Figures and Tables

**Figure 1 children-09-01208-f001:**
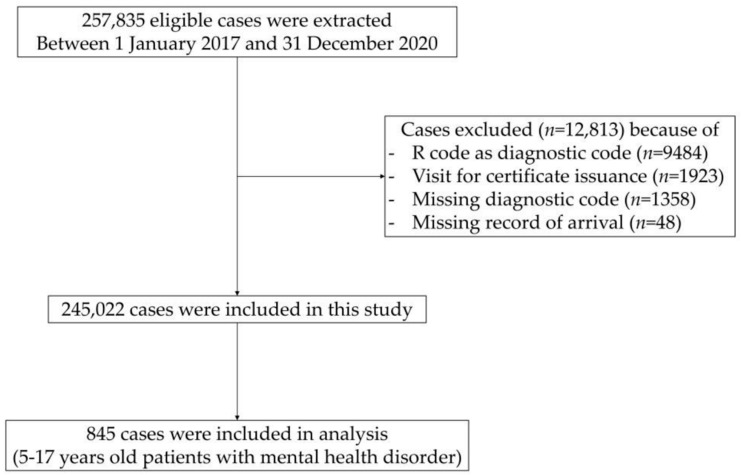
Flowchart of the study population.

**Figure 2 children-09-01208-f002:**
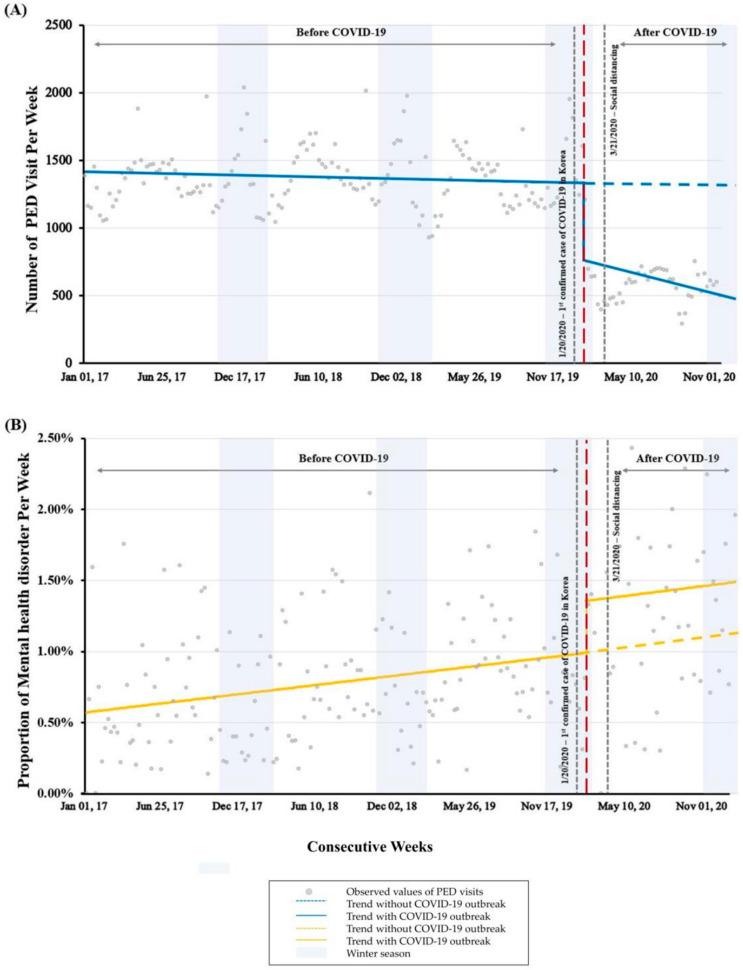
Segmented regression analysis of patients visiting the PED. Compared to before the COVID-19 pandemic, the weekly number of PED visits (**A**) have decreased significantly (*p* < 0.001) and the proportion of patients with mental illness (**B**) have increased significantly (*p* = 0.03). PED, pediatric emergency department. COVID-19, coronavirus disease 2019.

**Figure 3 children-09-01208-f003:**
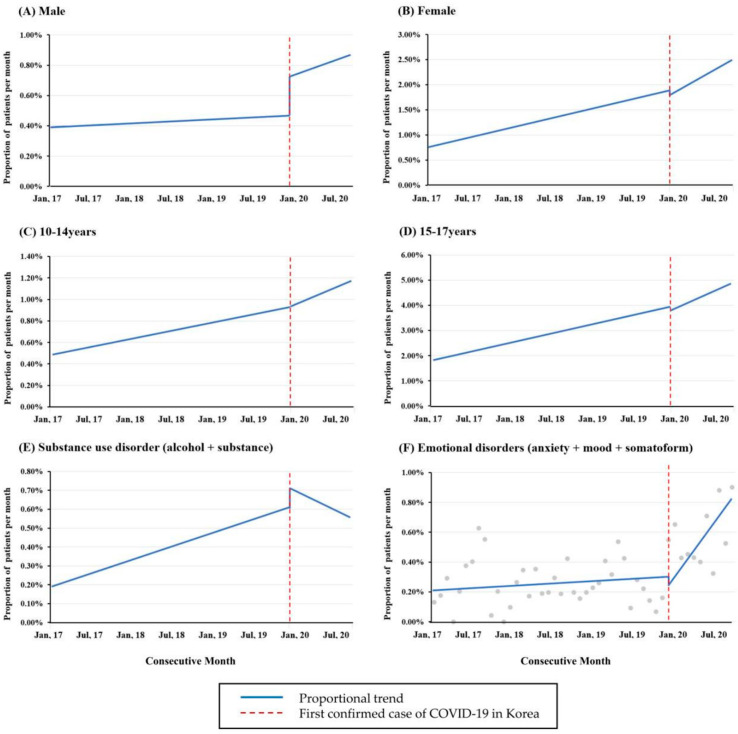
The segmented regression analysis of the monthly proportion of subgroups of patients with mental illness. (**A**) Male, (**B**) Female, (**C**) 10–14 years, (**D**) 15–17 years, (**E**) substance use disorder, and (**F**) emotional disorders. The changes in the proportion of male, female, 10–14 years group, 15–17 years group, and substance use disorders before and after the COVID-19 pandemic were not statistically significant. The proportion of patients with emotional disorders increased significantly by 0.06% per month during the COVID-19 pandemic (*p* < 0.001). COVID-19, coronavirus disease 2019.

**Table 1 children-09-01208-t001:** Characteristics of 5–17-year-old patients with mental illness in the pediatric emergency department.

Variables	2017–2019	2020	*p*
Total PED visits	84,790	14,883	NA
Total mental illness patient visits	657 (0.77)	188 (1.26)	<0.001 ^a^
Weekly visits, mean ± SD	4.2 ± 2.4	3.6 ± 1.8	0.05 ^b^
Age			
5–9 years	48 (7.31)	12 (6.83)	0.55 ^a^
10–14 years	172 (26.18)	43 (22.87)	
15–17 years	437 (66.51)	133 (70.74)	
Sex			
Female	450 (68.49)	130 (69.15)	0.93 ^a^
KTAS level			
1	4 (0.61)	3 (1.60)	0.08 ^a^
2	41 (6.24)	8 (4.26)	
3	262 (39.88)	93 (49.47)	
4	321 (48.86)	78 (41.49)	
5	29 (4.41)	6 (3.19)	
Mode of arrival			
Self-referred	607 (92.39)	170 (90.43)	0.25 ^a^
Referred from clinic	43 (6.54)	13 (6.91)	
Outpatient department	7 (1.07)	5 (2.66)	
Disposition			
Admission	122 (18.57)	33 (17.55)	0.83 ^a^

Note: All values are numbers (percentage, %) except where otherwise indicated. ^a^ Chi-squared test was used. ^b^ *t*-test was used. PED, pediatric emergency department. NA, not applicable. KTAS, Korean Triage and Acuity Scale. SD, standard deviation.

**Table 2 children-09-01208-t002:** The proportion of 5–17-year-old patients with mental illness visiting the pediatric emergency department.

Variables	2017–2019 (N = 84,790)	2020 (N = 14,833)	*p*
Total mental illness	657 (0.77)	188 (1.26)	<0.001
Anxiety disorders	100 (0.12)	32 (0.22)	<0.05
Mood disorders	39 (0.05)	17 (0.11)	<0.05
Somatoform disorders	73 (0.09)	24 (0.16)	<0.05
Developmental disorders	15 (0.02)	6 (0.04)	0.078
Schizophrenia and other psychotic disorders	14 (0.02)	3 (0.02)	0.749
Alcohol-related disorders	60 (0.07)	10 (0.07)	0.887
Substance-related disorders	265 (0.31)	78 (0.53)	<0.001
Intentional self-harm	28 (0.03)	8 (0.05)	0.216
Miscellaneous	63 (0.07)	10 (0.07)	0.775

Note: All values are numbers (percentage, %). *p*-values from the Chi-squared test.

## Data Availability

Data are available upon request owing to restrictions, such as privacy or ethics.

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
