# Peer review of "Impact of COVID-19 Pandemic on Children Visiting Emergency Department for Mental Illness: A Multicenter Database Analysis from Korea"

_children, 2022, doi:10.3390/children9081208_

Round 1

Reviewer 1 Report

A relevant topic has been appropriately addressed. It would be interesting to know if the relative increase in mental illness is due to a decrease in other causes of emergency visits. It is well known that infectious diseases (other than covid) and trauma decreased significantly during the lockdown / social distancing. Kindly include this information. 

It would complete the study if the authors can provide data along the same lines, after the resumption of normalcy after the pandemic. Did the trend return to the pre-covid levels? 

Author Response

Reviewer 1 Comments

Thank you for taking the time to review our manuscript and for providing meaningful comments. We have made corrections and clarifications in the revised manuscript after carefully reading your comments.

A relevant topic has been appropriately addressed. It would be interesting to know if the relative increase in mental illness is due to a decrease in other causes of emergency visits. It is well known that infectious diseases (other than covid) and trauma decreased significantly during the lockdown / social distancing. Kindly include this information. 

Response: Thank you for pointing this out. What you said is described in the discussion section (lines 190-198). In the discussion section, we explained that one of the reasons for the increase in the proportion of visits to children with mental illness was a significant decrease in the number of patients with infectious diseases and trauma.

lines 185-192: Another reason is thought to be a decrease in non-mental illnesses. Due to social distancing and strict personal hygiene during the COVID-19 pandemic, the prevalence of infectious diseases, particularly respiratory infections have decreased. A study in Singapore reported that respiratory infections were significantly reduced during lock-down due to the COVID-19 pandemic compared to the pre-lockdown period [20]. Additionally, due to limited outdoor engagement and activities, the number of injured patients has also decreased. An Irish study reported the fewest traumatic hospitalizations for children during the COVID-19 pandemic over the past decade [21].

It would complete the study if the authors can provide data along the same lines, after the resumption of normalcy after the pandemic. Did the trend return to the pre-covid levels? 

Response: Thank you for your important note. However, it will take weeks for us to have data on pediatric patients visiting emergency departments after social distancing is relaxed. Therefore, it is difficult to identify the data within the deadline for revision this time. In the future, we can conduct a follow-up study to identify the trend of pediatric emergency department visits among pediatric patients after the COVID-19 pandemic.

Reviewer 2 Report

Many thanks for inviting me to review this interesting paper. The study sought to examine the changes in the proportion of visits mental health visits to Emergency Departments made by children (aged 5-17) before and during the COVID-19 pandemic (2017 – 2020) in Korea. The authors of this study have produced a clear yet detailed manuscript, utilizing a comprehensive administrative data resource. There are some minor the authors may wish to consider clarifying in order to aid reading and interpretation:

·       The introduction could benefit from a broader discussion of the longitudinal research that has been conducted examining mental health during the first year of the pandemic. Although the authors have briefly mentioned some cross-sectional research (lines 37-42), a discussion of longitudinal studies, both comparing prevalence of mental health issues pre-pandemic and post-pandemic, and those examining changes in mental health during the pandemic would provide helpful context for the reader.

·       The authors state that: “…since the data of this study was provided anonymously, the possibility that information for the same patient might be included more than once cannot be excluded.” (lines 241 – 242). Does this mean that the dataset captured number of visits emergency department overall, rather than number of unique patients who attended the PED? If this is the case, the authors may wish to clarify wording in some cases. For example:

“We aimed to identify changes in children with mental illness who visited the pediatric emergency department (PED) during the coronavirus disease 2019 (COVID-19) pandemic.” (lines 15-16) – Should this instead state that the aim was 'to identify changes in the proportion of visits to the PED for mental health issues'?

“Our study showed that the proportion of total patients with mental illness visiting PEDs in Korea increased during the COVID-19 pandemic compared to that before the pandemic” (lines 164 – 166) – should this instead refer to the ‘proportion of total visits to the PED made by children for mental health reasons increased’, etc.?

·       “Those who visited the PED for issuance of a certificate or were discharged from the PED with R-codes indicating symptoms or signs as diagnostic codes and those with missing data during the intake process were excluded.” (lines 68-70) - It would be helpful to provide more context/information on what ‘issuance of a certificate’ and ‘R-codes indicating symptoms or signs as diagnostic codes’ refer to.

·       The discussion section may benefit from further consideration of the factors driving the increase in mental health related PED visits, over and above the stress associated with the pandemic. For example, before the pandemic, children and young people experiencing psychological distress may have accessed GP services, crisis intervention services and community and voluntary mental health groups for support. However, during the pandemic, these services may not have been available and therefore, in a time of distress, the only place they (or their parents/guardians) could access was the PED.

·       “Subgroup analyses revealed that emotional disorders significantly by 0.06% per month (95% CI: 0.02% to 0.09%, P < 0.001) during the pandemic.” (lines 24-25) – significantly increased?

·       “Moreover, as the reopening of schools is delayed and out-side activities are refrained, communication between children has decreased, and the time spent alone has increased.” (lines 48-49) – Is there a reference for this?

·       “during the one-year COVID-19 pandemic” (lines 170 -171) – ‘first year’ rather than ‘one year’ would be more appropriate here.

·       “This is probably because females experience more frequent  household mental illness…” (line 207) – what does ‘household mental illness’ refer to here?

·       “The heightened stress caused by the restriction of outdoor activities and friction with family members eventually stimulates emotional confusion and impulsivity in adolescents, which ultimately leads to substance use and abuse” (lines 235 – 237) – This sentence would benefit from rephrasing to clarify that stress may lead to emotional confusion and impulsivity, which in turn may lead to substance use in young people.

·       Online supplemental table A1 did not appear to be attached to the manuscript.

·       Figure 3 (F)  - typo ‘anxiety’

·       “This section may be divided by subheadings. It should provide a concise and precise description of the experimental results, their interpretation, as well as the experimental conclusions that can be drawn.” (lines 159-161) – this an be removed.

Author Response

Reviewer 2 Comments

Many thanks for inviting me to review this interesting paper. The study sought to examine the changes in the proportion of visits mental health visits to Emergency Departments made by children (aged 5-17) before and during the COVID-19 pandemic (2017 – 2020) in Korea. The authors of this study have produced a clear yet detailed manuscript, utilizing a comprehensive administrative data resource. There are some minor the authors may wish to consider clarifying in order to aid reading and interpretation:

Thank you for taking the time to review our manuscript and for providing meaningful comments. We have made corrections and clarifications in the revised manuscript after carefully reading your comments.

1. The introduction could benefit from a broader discussion of the longitudinal research that has been conducted examining mental health during the first year of the pandemic. Although the authors have briefly mentioned some cross-sectional research (lines 37-42), a discussion of longitudinal studies, both comparing prevalence of mental health issues pre-pandemic and post-pandemic, and those examining changes in mental health during the pandemic would provide helpful context for the reader.

Response: Thank you for your important. In addition to cross-sectional studies, longitudinal studies will also have an important meaning for readers to understand this study. Attached below is a longitudinal study reporting on depression and anxiety in children and adolescents before and during the COVID-19 pandemic. This study reported that the proportion of children adolescents with anxiety and depressive symptoms during the COVID-19 pandemic significantly increased from 14.9% to 30.1% and 10.0% to 15.1%, respectively, compared to before the pandemic. Added the following to the manuscript.

Line 45-48: In addition, a longitudinal study reported that the proportion of children and adolescents with anxiety and depression during the COVID-19 pandemic significantly increased from 14.9% to 30.1% and 10.0% to 15.1%, respectively, compared to before the COVID-19 pandemic [7].

[7] Ravens-Sieberer, U.; Kaman, A.; Erhart, M.; Otto, C.; Devine, J.; Löffler, C.; Hurrelmann, K.; Bullinger, M.; Barkmann, C.; Siegel, N.A.; et al. Quality of life and mental health in children and adolescents during the first year of the COVID-19 pandemic: results of a two-wave nationwide population-based study. Eur Child Adolesc Psychiatry 2021, 1-14, doi:10.1007/s00787-021-01889-1.

2. The authors state that: “…since the data of this study was provided anonymously, the possibility that information for the same patient might be included more than once cannot be excluded.” (lines 241 – 242). Does this mean that the dataset captured number of visits emergency department overall, rather than number of unique patients who attended the PED? If this is the case, the authors may wish to clarify wording in some cases. For example:

“We aimed to identify changes in children with mental illness who visited the pediatric emergency department (PED) during the coronavirus disease 2019 (COVID-19) pandemic.” (lines 15-16) – Should this instead state that the aim was 'to identify changes in the proportion of visits to the PED for mental health issues'?

“Our study showed that the proportion of total patients with mental illness visiting PEDs in Korea increased during the COVID-19 pandemic compared to that before the pandemic” (lines 164 – 166) – should this instead refer to the ‘proportion of total visits to the PED made by children for mental health reasons increased’, etc.?

Response: Thank you for your important note of what we were missing. As you said, my intention was to mean a change in the proportion of visits, not a change in the number of unique patient visits to the PED. It was revised as follows.

Line 15-16: We aimed to identify changes in the proportion of pediatric emergency department (PED) visits due to mental illness during the coronavirus disease 2019 (COVID-19) pandemic.

Line 159-160: This study showed that the proportion of total PED visits for mental illness in Korea increased during the COVID-19 pandemic compared to that before the pandemic.

3. “Those who visited the PED for issuance of a certificate or were discharged from the PED with R-codes indicating symptoms or signs as diagnostic codes and those with missing data during the intake process were excluded.” (lines 68-70) - It would be helpful to provide more context/information on what ‘issuance of a certificate’ and ‘R-codes indicating symptoms or signs as diagnostic codes’ refer to.

Response: Thank you for pointing out parts that may be difficult for readers to understand. An 'issuance of a certificate' refers to a patient who visited the PED to receive a document proving that he or she received treatment. And 'R-code' is a code starting with R in ICD-10 and indicates symptoms, signs or abnormal clinical and laboratory findings that are not classified otherwise. The manuscript has been revised as follows.

Line 68-72: Those who visited the PED for issuance of a certificate or were discharged from the PED with R-codes and those without proper data were excluded. An issuance of a certificate refers to a patient who visited the PED to receive a document proving that he or she received treatment. R-code is a code starting with R in ICD-10 and indicates symptoms, signs or abnormal findings that are not classified otherwise.

4. The discussion section may benefit from further consideration of the factors driving the increase in mental health related PED visits, over and above the stress associated with the pandemic. For example, before the pandemic, children and young people experiencing psychological distress may have accessed GP services, crisis intervention services and community and voluntary mental health groups for support. However, during the pandemic, these services may not have been available and therefore, in a time of distress, the only place they (or their parents/guardians) could access was the PED.

Response: Thank you for your important note of what we were missing. Due to social distancing during the COVID-19 pandemic, outside movement or gatherings were restricted, which made it difficult for mentally ill patients to use the primary care and counseling center. This reason is thought to have increased the proportion of PED visits among pediatric patients with mental illness. The manuscript has been revised as follows.

Line192-198: Restrictions on access to primary care may be another reason. Due to social distancing during the COVID-19 pandemic, outdoor activities or gatherings were restricted, which made it difficult to use primary clinics and counseling centers. Therefore, when a patient with a mental illness is experiencing psychological distress, the patient and their caregiver have no choice but to visit the PED, which may increase the proportion of patients with mental illness.

5. “Subgroup analyses revealed that emotional disorders significantly by 0.06% per month (95% CI: 0.02% to 0.09%, P < 0.001) during the pandemic.” (lines 24-25) – significantly increased?

Response: Thanks for pointing out where I made a mistake. It has been revised as follows.

Line 24-25: Subgroup analyses revealed that emotional disorders significantly increased by 0.06% per month (95% CI: 0.02% to 0.09%, P < 0.001) during the pandemic.

6. “Moreover, as the reopening of schools is delayed and out-side activities are refrained, communication between children has decreased, and the time spent alone has increased.” (lines 48-49) – Is there a reference for this?

Response: Thank you for your important note. The following references report that as school closures persist during the COVID-19 pandemic, mental stress, such as depression and anxiety, increases as adolescents spend less time playing sports with friends and more time at home. The following references are cited in the manuscript.

Line52-54: Moreover, as the reopening of schools is delayed and outside activities are refrained, communication between children has decreased, and the time spent alone has increased [8].

[8] Tang, S.; Xiang, M.; Cheung, T.; Xiang, Y.-T. Mental health and its correlates among children and adolescents during COVID-19 school closure: The importance of parent-child discussion. Journal of Affective Disorders 2021, 279, 353-360, doi:https://doi.org/10.1016/j.jad.2020.10.016.

7. “during the one-year COVID-19 pandemic” (lines 170 -171) – ‘first year’ rather than ‘one year’ would be more appropriate here.

Response: Thank you for your important note. Changing what you said better expresses the intent of our study.

Line 165-166: The strength of our study is that it was analyzed using a large data set extracted from six university hospitals.

8. “This is probably because females experience more frequent  household mental illness…” (line 207) – what does ‘household mental illness’ refer to here?

Response: Thank you for your important note. I should have stated the meaning of the sentence more concisely and clearly. I intended to state that females are more likely to have mental illness due to problems in the home because they experience more parental separation or divorce, challenges, and domestic violence. Household mental illness refers to mental illness caused by problems occurring at home. The content of the manuscript has been clearly revised as follows.

Line 207-208: This is probably because females experience more frequent parental separation and divorce, challenges, and violence at home during childhood than males.

9. “The heightened stress caused by the restriction of outdoor activities and friction with family members eventually stimulates emotional confusion and impulsivity in adolescents, which ultimately leads to substance use and abuse” (lines 235 – 237) – This sentence would benefit from rephrasing to clarify that stress maylead to emotional confusion and impulsivity, which in turn maylead to substance use in young people.

Response: Thanks for the sharp point. As you said, I have revised it to make the meaning of the sentence more precise and concise. The manuscript has been revised as follows.

Line 235-237: Restrictions of outdoor activities and friction with family can heighten stress, which may lead to emotional confusion and impulsivity in adolescents, which may ultimately lead to substance abuse.

10. Online supplemental table A1 did not appear to be attached to the manuscript.

Response: Thank you for your important note of what we were missing. Online supplemental table A1 is attached. "Please see the attachment"

11. Figure 3 (F)  - typo ‘anxiety’

Response: Thanks for pointing out where I made a mistake. I revised the figure 3 (F) as you said.

12. “This section may be divided by subheadings. It should provide a concise and precise description of the experimental results, their interpretation, as well as the experimental conclusions that can be drawn.” (lines 159-161) – this an be removed.

Response: Thanks for pointing out where I made a mistake. I have deleted the part you mentioned.
